# Like Riding a Roller Coaster: University Teachers’ Emotional Experiences Dealing with Student Plagiarism

**DOI:** 10.3390/ijerph20043276

**Published:** 2023-02-13

**Authors:** Xiaojing Wang

**Affiliations:** School of Foreign Languages and Literature, Beijing Normal University, Beijing 100875, China; queen8271@bnu.edu.cn

**Keywords:** plagiarism, academic writing, teacher’s emotion, higher education

## Abstract

The topic of research ethics has attracted attention in Chinese higher education, especially with regard to the “crackdown” on plagiarism. Although higher education teachers have implemented various interventions intended to reduce such misconduct, academic malpractice continues to occur. However, few studies have investigated the emotional challenges that these teachers face when dealing with plagiarism and the emotional changes that they experience in the process of addressing such misconduct. To address this research gap, in the current study, interviews, focus group discussions, and teaching journals were employed to investigate the negative emotional experiences of university teachers with student plagiarism in the Chinese context. An inductive thematic analysis was conducted, followed by in-depth analyses. The findings highlighted the fluctuating emotional development of the participating teachers from an ecological perspective, and the influential factors associated with the mitigation of negative emotions for teachers facing challenging situations were examined. The results also highlighted the necessity of taking the initiative to strengthen and normalize academic integrity in tertiary institutions.

## 1. Introduction

In recent years, an increasing number of studies have investigated the perceptions of different stakeholders regarding what constitutes plagiarism [1], various culturally and disciplinarily shaped attitudes toward plagiarism [2,3], and the pedagogical interventions that are employed to prevent plagiaristic practices [4]. These findings are supposed to improve teachers’ knowledge of the illegal reuse of text and provide them with effective measures to deal with any such cases appropriately. In fact, teachers may still experience various forms of distress when confronted with students’ acts of dishonest rule-breaking in the academic community [5,6]. More importantly, such emotional experiences could obviously impact their teaching behaviors and teaching quality, as well as teacher-student relationships and changes in education [7].

Research has indicated that emotions are related to cognition, thus suggesting that emotional reactions to difficult situations can impede an individual’s acquisition of knowledge and development of professional skills [8]. However, little evidence has been found regarding the authentic emotional experiences of teachers in the process of coping with students’ misconduct in the context of academic writing or the possible emotional changes that may be embedded in various “sociopolitical, institutional, cultural, personal, and interactional ecologies” [9]. In this case, it is important to investigate the “cracks” that university teachers face in the process of addressing instances of plagiarism and the emotional changes that may occur in this context. Hopefully, the findings of this research can contribute to the literature by allowing us to understand and optimize teachers’ emotions with the aim of deterring illicit text borrowing and providing more psychological support for teachers in their interventions against academic dishonesty.

## 2. The Literature Review

### 2.1. Conceptualizing Plagiarism

In higher education, “a well-integrated academic text should demonstrate thoughtful evaluation of the source texts through deep reading and logical structuring” [10]. In contrast, plagiarism indicates disrespect for authorship and critical thinking and illustrates the moral dilemma faced by university students. It has been found that the concept of plagiarism is difficult for many students to understand since sanctions for plagiaristic behaviors may be applied inconsistently, and teachers may have differing expectations regarding academic standards [5]. Teachers’ comprehension of plagiarism varies from a conception of moral decline or cultural clash to a lack of academic knowledge or even a combination of these factors [4].

Although definitional clarity is hard to achieve in this context, most scholars researching this topic have reached a consensus on the need to emphasize the “discourse of morality” when conceptualizing plagiarism [11]. This approach indicates the importance of ethical and intellectual concerns in academia. In the current study, plagiarism was, therefore, considered to be a form of academic misconduct that involves appropriating a range of materials or stitching sources together after making only superficial changes [11,12].

### 2.2. Teacher’s Cognition

Teacher cognition is a theoretical construct that is based on teachers’ experiences, and that highlights their mental complexity. This construct can be defined simply as the hidden cognitive dimension of teaching, which influences teachers’ educational decisions and practices [13]. In this context, teachers’ cognition regarding students’ unethical text reuse is also reflected in their teaching practices and influences students’ use of sources [14]. Investigation into how teachers perceive plagiaristic misconduct provides ideas and knowledge regarding how they present legitimate intertextuality to students and how they promote academic integrity among students [15].

Previous studies have shown that teachers perceive inconsistency with regard to recognizing and deterring plagiarism [5,15,16]. Although many teachers insist that all acts of copying constitute plagiarism and should be penalized [1], it is, nevertheless, important to distinguish “good” plagiarism from “bad” plagiarism before taking any punitive actions [17]. In simple terms, “bad” plagiarism is defined as deliberate copying without acknowledgment, which is an obvious rule-breaking behavior that receives little tolerance, while “good” or “acceptable” plagiarism is viewed as unintentional borrowing that occurs during students’ developmental stages [1]. In certain contexts, students are encouraged to borrow and copy good sentences from sources to enhance their own written work during a developmental stage. There is no denying that some teachers indeed view the memorization and imitation of the original literature as a learning strategy or literacy practice that can be employed for second language (L2) mediation [15,17].

In general, teacher cognition regarding students’ illicit text copying is discipline-based, culturally constructed, and experience-oriented [5]. First and foremost, teachers in hard disciplines, such as the natural sciences and engineering, demand stricter academic knowledge of plagiarism and exhibit less tolerance for unattributed discourse reuse in scholarly activities than do teachers in soft subjects, such as the humanities and the social sciences [2]. Such differences could be attributed to varying epistemological assumptions between soft and hard disciplines [18] and the distinct conceptions of textual ownership in different disciplines [19,20]. Moreover, teachers’ understanding of plagiaristic behaviors could also be viewed as “a cultural invention” [21]. Anglophone cultures usually strongly emphasize academic integrity [22], while non-Anglophone cultures advocate open and broad access to knowledge. In this context, unintentional rule-breaking through text reuse has received insufficient attention in non-Anglophone educational settings, and teachers with a collective orientation are likely to perceive intellectual products as the creation of the whole society [23]. However, enculturation may occur. For example, teachers trained in Anglophone cultures exhibit greater rigor and a more uniform understanding of textual misappropriation than teachers trained outside this context [24]. It has also been proven that teachers’ cognitions are not static. Teachers may learn from another culture and assimilate its practices and values. More importantly, some studies have reported that most countries that share the same cultural values occasionally fail to share the same perceptions of plagiarism [14,20,24]. In this case, “a simplistic cultural conditioning view” [3] should be rejected. In addition, teachers’ teaching experience has an impact on their understanding of transgressive intertextuality [25]. Researchers have highlighted a counterintuitive association between teachers’ professional experience and their perceptions of student plagiarism. Teachers who have taught for longer periods of time are more tolerant of plagiarism [15]. This tolerance may be attributed to lenient academic norms, according to which text copying is viewed as a language developmental strategy that represents students’ efforts to learn and practice [23]. Greater exposure to such standards can thus affect teachers’ stances regarding the norms of academic discourse [26].

Overall, regardless of whether plagiarism is good or bad, most teachers insist that all unethical practices in students’ academic work should ultimately be eliminated.

### 2.3. Teacher’s Emotions

The literature has proven that teachers’ cognition is directly correlated with their emotions [27], as “experiencing something occurs in the sphere of emotion, and its processing occurs in the sphere of cognition” [28]. Accordingly, due to teachers’ inconsistent cognitions regarding student plagiarism, they may experience some degree of serious emotional changes.

Primarily, a teacher’s emotion is viewed as a psychological construct consisting of internal feelings. Accordingly, following social constructionist approaches, teacher’s emotion refers to skill-based and socially constructed ways of being, which involve beliefs in the context of person-environment transactions instead of existing in an independent environment [29,30]. This definition reflects teachers’ dynamic perceptions of interactions and relationships within social and professional contexts at multiple levels, including both internal and external levels [7,31].

The dependency of the teacher’s emotion on the surrounding context could become manifest in interpersonal processes. In teaching settings, teachers’ emotions can be evaluated based on multilayered interactions [30]. When assessing students’ written work, teachers are more pleased when their students succeed in demonstrating expertise in academic discourse. This success thus enhances teacher wellbeing. This dependency of the teacher’s emotion on context evolves over time. When teachers experience clashes between their professional ethics and feeling rules regarding students’ written work, they are likely to face moral dilemmas and experience emotional labor [6,32]. For instance, when teachers fail to stop students’ academic misconduct, their major teaching rewards are disturbed. They then struggle with the tension between students’ negative responses and their professional responsibility. Once teachers come to view their devotion to their students’ work to be unrewarding, time-consuming, and insufficient, they enter a state of frustration, anxiety, and uncertainty [32,33].

A great deal of research has shown that teachers’ cognition, motivation, and behaviors are all impacted by their emotions and, in turn, shape student learning [30]. Above all, if teachers experience a sense of disbelief in their students’ academic integrity, they tend to interpret their students’ text appropriation as involving dishonest intent, and this distrust destroys student-teacher rapport over the long term [31]. In addition to its influence on teacher cognition, teachers’ motivation may also be affected by their emotions. In some cases, a teacher’s lack of enjoyment at work, particularly when they face confrontational situations in which their students exhibit hostile responses to their judgments, may lead to a decrease in motivation. Demotivated teachers may either become more lenient regarding students’ text misuse or attempt to stigmatize their students as exhibiting poor academic literacy [34]. Moreover, teachers’ negative emotions may cause them to become vulnerable in their teaching practice [30]. According to Vehviläinen et al. [6], some teachers’ reluctant treatment of student plagiarism is the result of their concern with the possibility of causing conflicts. The flawed practices of these teachers in response to illicit acts in the context of academic writing seriously impact their students’ attitudes toward their academic studies. Obviously, teachers’ emotions operate in the context of complex social interactions, and teachers’ affective experiences are, therefore, tied directly to their students’ learning outcomes.

In recent decades, an increasing number of studies have been conducted worldwide to explore teachers’ emotions and the factors that influence them from a multidimensional perspective [7]. Some such studies have reported that Chinese teachers usually attempt to maintain positive emotions at work and adopt various emotion regulation strategies to control their negative emotions with the aim of fulfilling their professional commitments [34,35]. Thus, in-depth research on Chinese teachers’ negative emotions seems to be lacking. This scarcity may be either methodology-oriented or context-oriented. To paint a full picture of Chinese teachers’ emotional feelings in a variety of complex contexts, further investigation should be conducted.

### 2.4. Theoretical Framework

To improve our understanding of teachers’ emotional development, the entire ecological system in which growth and change occur should be considered [36]. Bronfenbrenner’s ecological systems theory could help facilitate the exploration of the psychology and practice of language teachers and to provide ecological pictures of different research contexts [7,37]. More importantly, this approach could help us gain “in-depth insight into the factors influencing [teachers’] various emotions” [29]. Cross and Hong [7], drawing on Bronfenbrenner’s framework, claim that “the myriad of influences that impact the individual’s growth and development” could be highlighted.

Theoretically, this ecological system consists of five tiers of socially organized subsystems, including the microsystem, mesosystem, exosystem, macrosystem, and chronosystem [38]. The microsystem comprises the innermost layer in the most immediate context, in which specific activities are examined in relation to students and their family members. The mesosystem connects multiple microsystems within an organizational context, such as colleagues in universities. As an extension of the mesosystem, the exosystem refers to the external settings in which a teacher does not engage in direct interaction and is related to the development of the individual, namely, the community setting. The macrosystem encompasses the broader political, cultural, and historical society in which the previous systems are embedded. The chronosystem includes the temporal events that transpire in an individual’s life at the outermost level, which reflect his or her life changes [38]. When using these five subsystems to explore teachers’ emotions, the evolving interactions between the teachers and their entire environment are identified explicitly, as is the way in which they manage these complex interactions.

In previous research on this topic, the ongoing emotional challenges faced by teachers in response to students’ illicit text reuse in the context of academic essays have been absent. More importantly, previous studies have failed to trace the emotional changes that occur in teachers in response to their students’ illicit acts, and the factors that contribute to such changes in a broader ecosystem remain unknown, with little qualitative evidence in the Chinese educational context. Therefore, the aim of the present study was to highlight this research need and fill this gap based on the nested structure of Bronfenbrenner’s ecological model. Thus, the following research questions were addressed in this study:Do Chinese teachers of English experience any emotional challenges in their interactions with ecological systems when addressing student plagiarism in the context of academic writing?Do any emotional changes occur during these interactions? If so, what factors affect the emotional changes that occur in teachers when viewed from an ecological perspective?

## 3. Research Design

### 3.1. Research Participants

The research participants were recruited using a convenience sampling method, beginning with the first author’s personal contacts [39]. The first author contacted teachers from five different universities located in mainland China who were potentially interested in participating in this research. Eleven participants were approached via email and subsequently informed of the research purpose and their rights through online conferencing. Ultimately, six teachers were selected to participate. The selection criteria were as follows: (1) working in comprehensive universities, including key national universities (Project 985 or Project 211) and ordinary universities; (2) having various personal, educational, and professional backgrounds; and (3) teaching writing during the spring 2022 semester, which allowed the participants to record their on-the-spot feelings with respect to students’ misconduct.

According to the participating teachers’ profiles, which are shown in Table 1, teachers in the EM group (i.e., those who taught English majors) all held doctorate degrees and had various levels of experience studying abroad. In these respects, they differed from teachers in the NEM group (i.e., those who taught non-English majors). This difference was the result of the different employment requirements for teaching English majors and teaching non-English majors. This difference has been decreased to some extent, but it continues to exist.

### 3.2. Research Methods

Following the suggestion of Yu et al. [30] that “qualitative and interpretive methodologies are required in order to gain access to the meanings of particular emotional experiences from an emic perspective”, a qualitative multicase method was employed in the current study. This approach could yield a deeper understanding of individual teachers’ emotions and ensure depth regarding the factors influencing those emotions [29]. Semistructured interviews, focus group discussions, and teachers’ journals were used to explore the emotional experiences of teachers as they negotiated the tensions associated with dealing with students’ plagiarism.

The semistructured interviews consisted of 11 questions pertaining to teachers’ beliefs regarding academic writing ethics and the emotional challenges that they experienced in responding to student plagiarism; these questions followed the work of Benesch [40], Davis and Morle [41], and Cheung et al. [42]. Subsequently, a focus group discussion was conducted to explore the following topics: (1) the emotions experienced by teachers when they noticed student plagiarism; (2) the emotions they experienced during teacher-student interactions regarding plagiarism; and (3) the emotional changes they exhibited in reaction to students’ follow-up responses to instances of plagiarism. This discussion was conducted to allow teachers to share their thoughts and express the emotions they experienced when dealing with plagiarism.

To “involve the least amount of researcher control” [43], participating teachers recorded their emotional responses to student plagiarism in teacher journals. The use of journals could not only prompt the teachers to reflect on their real experiences when grading academic essays but could also help them express their on-the-spot emotions and thoughts regarding students’ text misuse. They were required to rate their feelings on a scale ranging from −5 as the lowest to 0 as the highest with regard to their negative emotions within different ecological systems (prior to data collection, appropriate training was provided in rating). Their journal numbers varied due to their different assignment arrangements. The teachers’ shared language, i.e., Chinese, was used in the interviews, focus group discussions, and journals to help them express their ideas explicitly in a relatively easy and relaxed manner.

### 3.3. Data Collection

Data were collected from March 2022 to June 2022. All the participating teachers were informed of the purpose and significance of this study. Consent forms were signed after the teachers confirmed their willingness to participate. The participants were interviewed individually in April and May 2022, and each interview lasted 30–35 min. The focus group discussions were conducted in June for the EM group and the NEM group. Each discussion lasted for approximately 45–50 min. Since the universities with which the participating teachers were affiliated are located in different provinces of China, both interviews and focus group discussions were conducted online via Zoom (known as Zoom Video Communications, is a technology company. Founded in 2011, Zoom is publicly traded and headquartered in San Jose, CA, USA) and recorded using EV (version 4.2.2, software for recording computer screens, founded in 2015, headquartered in Changsha, Hunan, China) with the participants’ permission. In addition, the participant teachers submitted their journals after grading their students’ final essays. Twenty-nine journal entries were collected. With the exception of T4, who assigned only four written works, the other five teachers wrote five journals since they assigned five essays. The total word count of the 29 journals was approximately 10,186 words.

### 3.4. Data Analysis

The interviews and focus group discussions were transcribed verbatim, resulting in a data corpus featuring a total of 89,648 characters. Inductive thematic analysis was conducted by reducing the extensive transcripts into core themes that reflected the overall context [37,44]. Following the suggestions of Gkonou and Miller [45], these transcripts were read multiple times, line by line, and labeled to “develop deep familiarity” (p. 143). This organic process facilitated the emergence of code nodes and the formation of different thematic categories in response to the research questions. Beginning with the main categories, salient remarks were labeled, and similar patterns were clustered within more abstract subcategories, including the teachers’ underlying cognitions, the emotional challenges they faced, and any changes that occurred in their emotions at different stages of the process of detecting and addressing plagiarism. To ensure the reliability of the codes, two research assistants worked independently. Their level of agreement regarding the coded work was high, i.e., their level of agreement was 91% [7]. All disagreements were further resolved with the help of the researchers. To counteract misinterpretations, the data were sent to the participants for review [46]. Ultimately, all data from the interviews, focus group discussions, and teachers’ journals were merged, modified, double-checked, and complied.

For instance, the participants’ statements regarding their emotional experiences were first labeled. Subsequently, the emerging codes were grouped to establish connections and compared both within and across cases [47], consolidated through axial coding, and later categorized under the subconstruct of selective coding. After each round of revision, the data were recoded until no new codes, categories, or themes emerged. Moreover, the relationships among each main category were identified through axial coding. In this way, the subset of data was sorted according to the identified nodes and subsequently assigned to 19 codes, which were later refined to produce 15 codes. All codes were categorized inductively in alignment with the research questions. Sample data coding is shown in Table 2.

### 3.5. Ethical Considerations

In this study, the participants could experience discomfort and ambivalent emotions since the teachers were required to report sensitive data that could be related to their workplace and colleagues. Following the approach used by Mercer [9], the data collection and analysis procedure involved a high degree of confidentiality to protect the interests of the participating teachers and other relevant parties.

### 3.6. Trustworthiness and Limitations

As this is a small-scale study, it is not possible to generalize the findings of this study to the larger population of Chinese teachers in all disciplines and from different provinces. However, university teachers usually share some common ground across disciplines with regard to the process of dealing with plagiarism. It is hoped that due to the detailed descriptions of these participants’ struggles with this issue, the findings of this study may be transferable to other settings [48]. Additionally, a certain degree of subjectivity may be involved in presenting and interpreting the data. However, this subjectivity was mitigated by the double examination of the self-reported data and the explicit explanation of the research purpose and demonstration of the researchers’ role [49]. In addition, teachers tend to hide their negative emotions in public [34]; hence, reconfirmation of data confidentiality was emphasized to reduce the risk of inauthentic responses. Due to the mutual trust developed between the authors and the participating teachers, the teachers were likely to express their sincere thoughts and reveal their real concerns about students’ plagiarism.

## 4. Findings

### 4.1. Emotional Challenges in Different Ecological Systems

#### 4.1.1. Microsystems

Based on their direct contact with students, the participating teachers reported various levels of difficulty with regard to the task of addressing academic misconduct. These difficulties caused the teachers to encounter various emotional challenges at the microsystem level.

**Extract 1**: It is truly frustrating when I see students’ unacceptable matching text. I have told them how to cite the source. It is very basic knowledge for a university student (T1, journal 2).

When faced with students’ abuse of text-borrowing rules, T1 and other teachers felt that their efforts in establishing academic standards in the class had been fruitless. Students’ passive responses, such as their indifference to the intervention regarding ethical principles, were a major factor associated with T1’s negative emotions.

Another frequently mentioned negative emotion was loneliness. These teachers are expected to establish connections with their colleagues in the context of addressing instances of text appropriation on the part of students. Unfortunately, they ultimately experienced an unpredictable sense of loneliness, as in the case of T6.

**Extract 2:** My colleague stuck to her own perceptions on dealing with students’ text copying. We had little discussion. I felt sad and lonely… (T6, journal 2).

In addition, T1, T2, T3, and T5 reported self-condemnation regarding their teaching competence. When they encountered frequent failures in their attempts to strengthen students’ academic integrity, they developed low self-esteem.

**Extract 3:** When exact block copies in students’ essays were not referenced, I experienced self-condemnation. I don’t know how to make my students aware of the ethical rules (T3, interview).

Teachers’ negative emotions were also elicited by their universities. Four teachers, including T4, as shown in Extract 4, expressed anxiety due to the lack of technology-enhanced software for the detection of source appropriation and to support their decisions regarding suspected text misuse. In addition, with the exception of T6, the other teachers, such as in Extract 5, expressed their dissatisfaction with the lack of any official emphasis on ethical principles on the part of the university.

**Extract 4:** I was anxious when addressing students’ text copying. Without text-matching software, I could barely decide whether their work was copied… My university didn’t provide such software (T4, journal 2).

**Extract 5:** Honestly, I do not know where to find the official ethical rules on the university website. I am so annoyed. I tried to ask other teachers, but they gave no definite answers (T5, journal 1).

As shown above, in the microsystem, the participating teachers experienced emotional challenges, which varied in terms of their students’ negative reactions to academic integrity, their colleagues’ divergent treatment of students’ misconduct, their lack of professional knowledge, and the lack of official support from their universities.

#### 4.1.2. Mesosystems

Various types of mesosystems emerged. One common mesosystem was composed of colleagues, the university, and the teacher. As noted by T2 in Extract 6, when he performed his duties and exhibited good ethical practice in teaching, his colleagues’ treatments of students’ literary theft would cause him to experience great pressure and aggrievement.

**Extract 6:** Some colleagues follow their own standards and show tolerance regarding the plagiarized text. There is no clear guidance for teachers; it made me feel aggrieved (T2, journal 3).

Ideally, teachers and their colleagues should share certain common values and follow university regulations in their teaching practices. However, in the case of T2, this unhealthy mesosystem was the result of his colleagues’ lenient attitudes and the university’s vague reinforcement of ethical standards.

In the teacher-student-colleague mesosystem, some colleagues’ excessive tolerance of students’ misuse of texts irritated the participating teachers. For example, T4 complained about his insistence on standard ethical codes.

**Extract 7:** My students complained about my harsh treatment of their plagiaristic behaviors, as other teachers were more tolerant of such conduct. I felt so mad (T4, interview).

In Extract 8, negative emotions emerged in the context of a teacher-student-university mesosystem. T4 and T6 were depressed since their strict decisions regarding students’ misconduct resulted in poor teacher evaluation grades or low module selection rates. These factors may affect a teacher’s professional progress. Although this issue may not be a common phenomenon, it, nevertheless, affects teachers’ professional decision-making.

**Extract 8:** T6: You know, my strict ethical decisions disgusted students. It may have led to low course enrollment. I was so stressed.

T4: My colleague failed two students due to plagiarism. It’s hard to say whether the two low marks in her end-of-term evaluation were given by them. It is somewhat possible.

The universities of T4 and T6 failed to employ official measures to protect teachers’ rights at this point. This failure, thus, damaged the teachers’ commitment to moral behavior in academic settings.

Overall, the mesosystem is integral to the changes that occur in teachers’ emotions. Various conflicts emerged; therefore, the teachers developed negative emotions within these weak mesosystems.

#### 4.1.3. Exosystems

Four types of exosystems were described by the participants. First, T2 and T3 expressed their demotivation resulting from their family member’s incomprehension of their strictness regarding students’ literary theft. For instance, the complaints that T2 received from his wife discouraged him from addressing students’ academic malpractice.

**Extract 9:** My wife used to suspect the values underlying my strict acts… I gradually became demotivated when dealing with students’ misconduct (T2, interview).

Moreover, T3, T4, and T6 reported their regrets regarding their treatment of students’ textual appropriation as a result of their changing identities. In the case of T3, her role change led to her failure to continue to follow the ethical rules she had established. Her identity-based decision-making regarding students’ academic malpractices was unconscious and shaped by her professional beliefs. This change may, in turn, have affected her social relationships with her colleagues. Thus, this exosystem influences the teacher-student-colleague mesosystem.

**Extract 10:** […] In front of my students, I could be their friend, their Agony Aunt, their teacher, their facilitator … my role-changing sometimes impacted my thoughts about sticking to the academic ethical rules. I felt very regretful (T3, journal 3).

In addition, the educational policy framework can impact certain mesosystems that impact teachers’ emotions. In recent years, the topic of academic integrity has received increasing emphasis in educational policy reinforcement. This change has impacted the ethical standards stipulated by universities. Thus, the teacher-university mesosystem has been affected.

**Extract 11:** Academic integrity has been emphasized by the Ministry of Education. My university also highlighted it but took few effective measures. It is annoying. (T1, interview).

Teachers’ emotions seemed to have little relation to the responses of students’ peers (i.e., students’ friends and acquaintances, whose impact on the decisions made by those students may be greater than that of their teachers). In fact, these teachers were impacted by these responses on the part of students’ peers, as students’ decisions regarding citing strategies could be influenced by their peers’ suggestions, as shown in Extract 12.

**Extract 12:** I became impatient as my students ignored my warnings about text copying. They said that their peers told them the copied texts were hard to identify. (T6, journal 4).

In general, the participating teachers experienced four major emotional challenges in the context of the exosystem, which resulted from their family environment, their various professional roles, educational policy, and the responses of students’ peers.

#### 4.1.4. Macrosystem

In the current setting, the macrosystem refers to the attitudes and ideologies associated with the sociocultural context in which teachers’ emotional challenges are shaped. T1, T2, and T5 expressed ambivalent emotions regarding students’ misuse of texts since they struggled to determine whether such misuse was the result of the collective culture, according to which memorization and imitation were ways of expanding the common pool of knowledge [50]. This emphasis has led to increased teacher empathy for students’ illicit behaviors.

**Extract 13:** I am kind of ambivalent about students’ literary misconducts since they may not understand why they should cite. It is a culture-related issue (T3, journal 3).

The case of T3 showed that the unsuccessful comprehension of academic integrity is the result of collective cultural norms. The shared value prioritized in this context is deeply rooted in some students’ minds, affecting the methods they use to acknowledge sources and triggering teachers’ negative emotions.

Overall, the negative emotions experienced by these teachers within different levels of ecological systems are illustrated explicitly in Figure 1 in accordance with the findings reported above. When making decisions regarding students’ academic misconduct, teachers’ negative emotions emerged during their interactions with the abovementioned environments.

### 4.2. Changes in the Emotional Challenges Faced by Teachers in the Chronosystem

A chronosystem comprises both normative and non-normative life transitions [38,51]. In the data referenced by this study, the transitions were normative, which was established based on the intervals of students’ written essays. Following the framework developed by Sun and Yang [29], the holistic development of the emotional challenges faced by individuals at different stages is shown in Figure 2, Figure 3, Figure 4, Figure 5, Figure 6 and Figure 7. Teachers graded their negative emotions at each interval. A score of −5 represents a totally negative emotion. The numbers between −5 and 0 represent different degrees of negative emotions.

As shown in Figure 2, T1’s negative emotions within the microsystem and macrosystem were distributed across a wide range. While grading essay 2, T1 became angry since she observed little improvement in text copying after her in-class intervention. These feelings were alleviated when she graded essay 3 when she noticed that the students’ false awareness of common knowledge was related to their cultural background. This turning point in the mesosystem resulted from the strengthened relationship between university support and students’ enhanced citing strategies. Within the exosystem, T1’s emotional challenges tended to reach a balance.

**Figure 2 ijerph-20-03276-f002:**
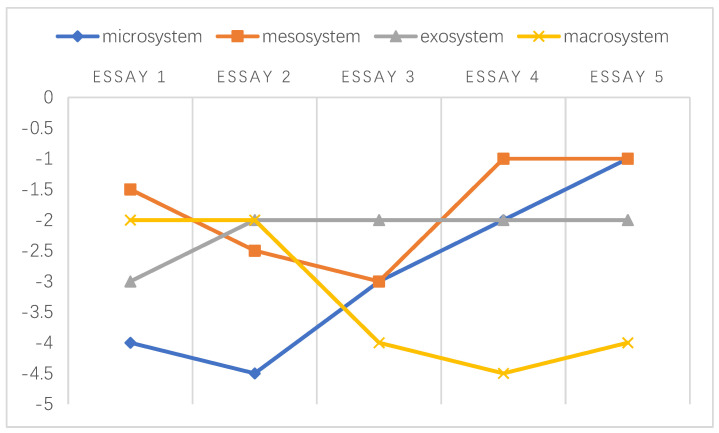
An illustration of T1’s emotional changes in the ecological systems after grading five essays.

The emotional changes experienced by T1 may have resulted from her individual background. Her negative emotions in the microsystem and macrosystem may be due to her overseas study experience. Since she was familiar with academic integrity in Western universities, she could barely tolerate textual abuse in written work, similar to the findings of Hu and Shen [24]. When students challenged her ethical perceptions, T1 exhibited low spirits.

**Figure 3 ijerph-20-03276-f003:**
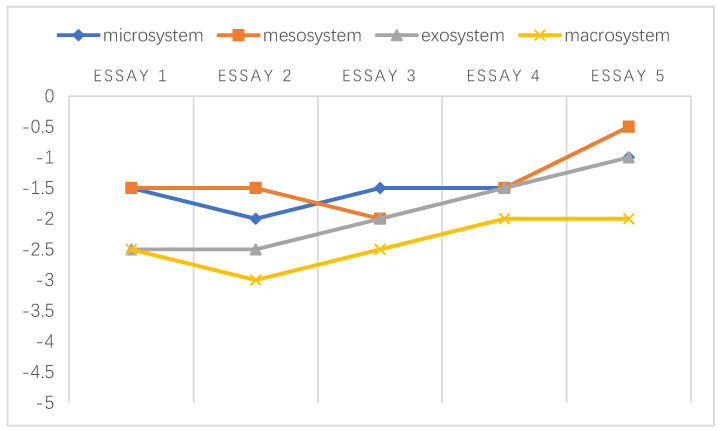
An illustration of T2’s emotional changes in the ecological systems after grading five essays.

T2’s negative emotional trajectory exhibited a uniform distribution in all systems. It seemed that T2’s ability to cope with emotional discomfort was relatively high; in other words, his emotional capacity helped him adapt to challenging situations. As shown in Figure 3, T2 exhibited encouraging emotional development in the exosystem since he found an effective way to overcome adversity and prevent himself from limiting his beliefs, as described in Extract 14.

**Extract 14**: The talk with my wife allowed me to see my professional aspirations clearly, and she supported my insistence on academic ethics (T2, journal 3).

**Figure 4 ijerph-20-03276-f004:**
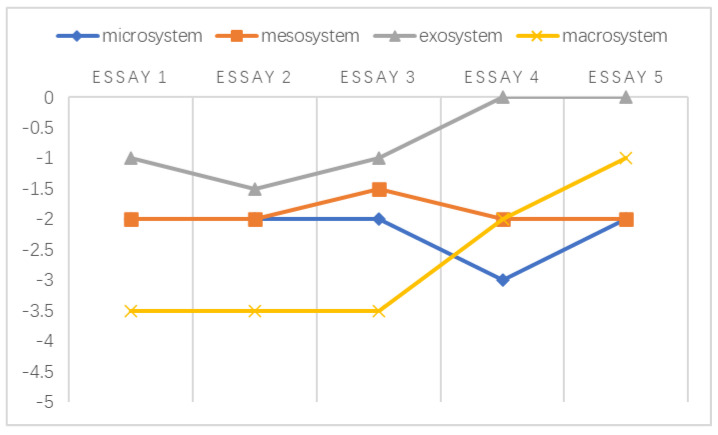
An illustration of T3’s emotional changes in the ecological systems after grading five essays.

As illustrated above in Figure 4, the negative emotions experienced by T3 within the mesosystem remained rather flat over time. Based on these findings, her turning point in the exosystem stemmed from her changing identities. She was more tolerant of students’ written misconduct when she became their facilitator. This turning point in the microsystem and macrosystem was related to a change in her professional beliefs. T3 previously attributed students’ textual appropriation to their sociocultural origins. During her in-service training, she came to understand that the key to this issue may be the students’ scope of knowledge, as described in Extract 15.

**Extract 15:** I agreed with the teacher trainer that the post-1990s generation might not be deeply influenced by the collective culture. Their text-copying problem was related to the constraints on their scope of knowledge (T3, journal 4).

**Figure 5 ijerph-20-03276-f005:**
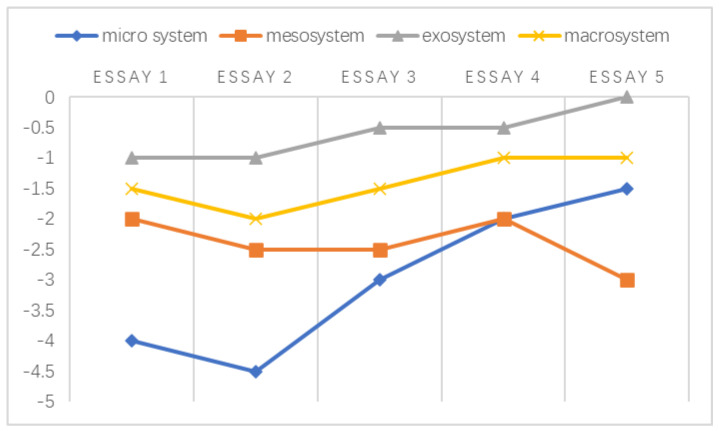
An illustration of T5’s emotional changes in the ecological systems after grading five essays.

According to Figure 5, the negative emotions experienced by T5 remained rather stable in the exosystem and the macrosystem. Instead, she developed a largely undulating emotional response in the microsystem when negotiating with students regarding their text appropriation. Her negative emotions decreased after essay 3. As a result of her newly designed paraphrase exercises, the students’ misuse of texts was gradually eliminated.

**Extract 16:** T5: Students’ insufficient language skills lead to incorrect text borrowing. Teaching them how to paraphrase is quite important.

T4: You are absolutely right.

T5: My designed exercises worked well for my students. That’s fantastic.

Surprisingly, T5’s emotions in the mesosystem failed to develop in a positive manner. The results indicated that the ineffective interaction between the university and the colleague microsystems affected her emotions. According to her, the ethical values that her university held conflicted with the values that her colleagues held regarding students’ academic misconduct. This weak mesosystem caused T5 to face long-term pressure at work.

**Figure 6 ijerph-20-03276-f006:**
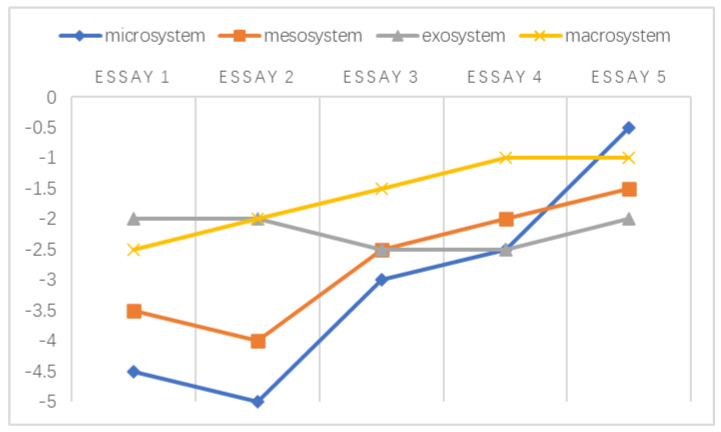
An illustration of T6’s emotional changes in the ecological systems after grading five essays.

A similar emotional trajectory occurred in T6’s microsystem and mesosystem in Figure 6. The negative emotions experienced by T6 were mitigated after essay 3. In the microsystem, T6 identified his divergent treatment of text misuse. To be fair to his students, he discussed this issue with his teaching group several times and ultimately achieved professional consistency with his colleagues. Moreover, he reported his concerns regarding teacher evaluations and course enrollment to the university, and a reformed policy (announced during the period of assigning essay 2) was developed to protect the interests of both teachers and students.

**Figure 7 ijerph-20-03276-f007:**
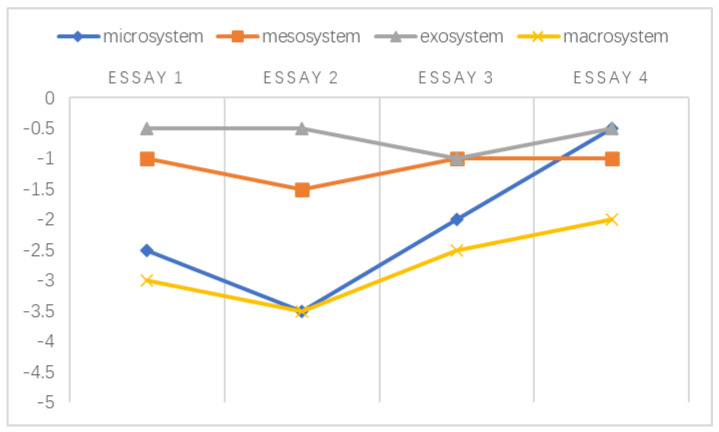
An illustration of T4’s emotional changes in the ecological systems after grading four essays.

In Figure 7, in addition to the insignificant changes in the mesosystem and exosystem, a sudden increase occurred within T4’s microsystem and macrosystem when grading essay 3. The factor that drove this emotional change in T4 was organizational support. Her school provided access to plagiarism detection software that could be used to assess essay relevance and reduce students’ academic appropriation. T4, thus, felt less anxious with regard to identifying students’ text misuse.

According to the six participants’ emotional changes in the chronosystem, in addition to the exceptional case of T5, all negative emotions gradually decreased over the course of the semester. The participating teachers strove to negotiate with students regarding their academic misconduct in the ecological systems in the process of addressing instances of plagiarism.

## 5. Discussion

The findings of this research have proven that Chinese teachers of English experience various types of emotional challenges in their interactions with ecological systems when addressing student plagiarism in the context of academic writing. As a result of the forceful efforts made by most participating teachers to eliminate passive emotions, a rather positive route of emotional development emerged in the ecological system, which was fraught with twists and turns. The major driving factors underlying such positive emotional development varied.

Within the microsystem and mesosystem, the major contributing factors include better-developed collegial relationships, increasing institutional emphasis on academic integrity, and teachers’ and students’ convergent expectations of text reuse. First, the participating teachers strove to develop collegial solidarity and recognition in the context of addressing students’ unethical behaviors. There is no denying the fact that teachers pursue individual excellence at work, even in the context of harmonious collaboration. In this case, the potential value of collegial relationships grounded in teachers’ professionalism became evident. According to T6, with the support of shared professional beliefs, a consensus was ultimately reached after repeated negotiations, and the teachers’ negative emotions were gradually mitigated. This finding is basically in line with Shah [52], whose study suggests that highly collegial cultures in higher education augment teachers’ professional growth, job satisfaction, as well as student performance. Additionally, the strengthened institutional support in enforcing academic honesty helped teachers conduct an easier assessment and brought them sufficient confidence in teaching practice. In the current study, four teachers mentioned that their universities either provided teacher training programs to improve their professional ability or offered access to plagiarism checker software to identify duplicate text use. Although some of these measures were only temporary, to some extent, they rectified rigorous deficiencies in teachers’ professional knowledge, increased their power, and raised their spirits at work. Another contributing factor was the convergent expectations of teachers and students concerning text reuse. All participating teachers experienced pronounced emotional challenges while grading the first few essays, indicating that their expectations regarding the citing of sources exhibited a mismatch with their students’ performances. This tension existed due to complex historical and pedagogical reasons. The turning point ultimately occurred when teachers, such as T5, identified the students’ limited scope of knowledge regarding text copying. T5 attempted to establish mutually respectful relationships between teachers and students and to employ effective interventions to help reduce text appropriation. The teachers felt encouraged by the increasing standardization of students with regard to the reuse of academic texts.

At the exosystem and macrosystem levels, the dominant factors that drive the emotional changes that occur in teachers in challenging situations include their expanded emotional capacity and improvements in teacher agency. On the one hand, in challenging situations, teachers with expanded emotional capacities could grow in a healthier way. Most teachers struggled to adjust their emotional responses when facing difficult circumstances. For instance, T2 was demotivated since his wife expressed incomprehension about his serious treatment of students’ plagiarism. Fortunately, such a passive emotion occurred for a rather short period in the case of T2. He proactively managed his negative emotions to prevent reaching their full emotional capacity. He made certain adjustments to obtain understanding from his wife through frequent communication. Such proactive communication was the outlet to allow his negative emotions to flow out. T2’s superior responsiveness to tough situations reflected his high emotional capacity. On the other hand, high-level teacher agency led to positive emotional development for the participating teachers. Teacher agency is practiced “when [teachers] make choices and take stances in ways that affect their professional positions” [53]. Thus, teachers who have high-level agency are more aware of their goals for professional growth and take positive actions. The degree to which a teacher acts with agency depends on both a teacher’s internal traits as well as a university’s structural conditions for professional development. According to T3, the in-service training arranged by her university corrected her misguided professional beliefs and subsequently established conditions that offered the teachers more autonomy and allowed them to engage in continuous professional development. Although T3 continued to suffer hardship with regard to growth, she felt empowered and took ownership of her teaching practices. In this way, her negative emotions became less pronounced and were unconsciously mitigated by her enhanced understanding of the sociocultural environment.

In addition to the abovementioned positive growth of the teachers’ emotions, T5’s reverse route of emotional development in the mesosystem aroused further attention. During the essay-grading period, T5 attempted to extricate herself from her negative emotions in the mesosystem but ultimately failed to do so. In essence, teachers’ beliefs about learning and teaching influenced their instructional decisions and, in turn, their emotions. The incorrectly reframed ethical policy remained ingrained in some teachers’ minds and, thus, clashed with the university’s rules for academic integrity. This conflict led to an unsound academic atmosphere, sent confusing signals to the stakeholders, including both teachers and students, and thus, evoked emotional stress in various settings. It is, therefore, suggested that effective measures and continuous meso-level interactions, such as continuing professional development programs for in-service teachers, could help transform teachers’ professional beliefs and promote positive emotions in the long term [36].

On the whole, within the chronosystem, most participating teachers strove to extricate themselves from emotional exhaustion across ecological systems. This study demonstrated that teachers move vigorously toward a consistent understanding of goodness in the context of academic writing; the goodness of students is in accordance with the goodness of their English teachers, organizational communities, and even society as a whole. Additionally, T5’s route of emotional development in the mesosystem not only revealed the need to reinforce institutional policies and promote continuous interactions in response to standardization but also highlighted the necessity for conducting a longer period of research.

## 6. Conclusions

From the perspective of qualitative design, the current study illustrated the ongoing emotional challenges that university teachers experienced when dealing with students’ illicit text copying from Bronfenbrenner’s ecological perspective, thus improving our understanding of the corresponding factors influencing teachers’ emotional development in higher education. The multiple cases referenced by this research highlighted the emotional changes that occurred in teachers in different ecological systems. In one exceptional case, the negative emotions of most teachers were mitigated during the essay-grading period. Such a positive transformation can be attributed to various factors, including better-developed collegial relationships, increasing institutional emphasis on academic integrity, the convergent expectations of teachers and students regarding text reuse, teachers’ expansion of their emotional capacity, and the enhancement of teacher agency. Extra attention is given to the regressive emotional change in the mesosystem. This case indicated the need for long-term research spanning a broad range of spatial and temporal scales.

Based on the findings, this study carries several important implications for L2 English teachers in higher education. First, given the students’ reactions to their illicit acts in academic writing, teachers should consider how to provide students with more effective interventions. For instance, the academic dishonesty policy can be explicitly stated in the syllabus to help reduce plagiarism through the “threat” of being caught and punished. Additionally, aligning with Hu and Shen [24], designing and providing appropriate and sufficient intertextual practices could facilitate students’ understanding of various patterns of text reuse. Second, teachers’ varied treatments of students’ plagiarism indicated the necessity of engaging in continuous professional development. It could not only enhance teachers’ professional competence but also extend their scope of knowledge regarding institutional and sociocultural ethical standards. More importantly, with expanded knowledge and enhanced skills, teachers could generate effective intertextual practices. Third, the findings suggest that there is a need for teachers to take the initiative in minimizing negative emotions at work. Seeking support from colleagues and local authorities, or developing social-emotional skills to rediscover the joy in teaching, can help teachers overcome their negative emotions and improve their emotional health, which, therefore, ensures a positive and safe teaching environment.

Future research can make use of quantitative data to paint a complete picture of the emotional changes that occur in teachers when they face various problems regarding academic integrity with the aim of obtaining a deeper understanding of the extent to which teachers’ emotions vary in different emotionally challenging situations. Additionally, more studies are necessary to explore the transition from a newly enrolled teacher to an experienced teacher with a focus on the dimension of the on-the-spot emotional changes that occur in teachers when they must address students’ malpractice. These studies could contribute to the strengthening and normalization of academic integrity in higher education, thus empowering teachers in the long run.

## Figures and Tables

**Figure 1 ijerph-20-03276-f001:**
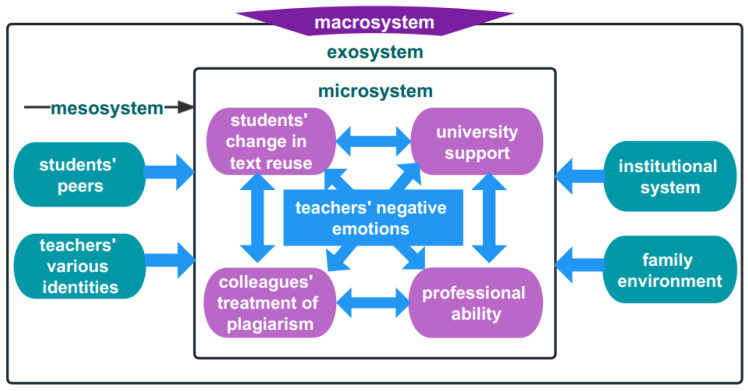
Teachers’ negative emotions within ecological systems.

**Table 1 ijerph-20-03276-t001:** Backgrounds of the Participating Teachers.

	Students Taught	Sex	Age	Degree	Years of Teaching	Years of Studying Abroad	University Level
T1	NEM (non-English majors)	F	29	Ph.D.	0.5	5	985/211
T2	NEM	M	44	Ph.D. candidate	10	0.5	/
T3	NEM	F	55	MA	31	0	211
T4	EM (English majors)	F	41	Ph.D.	10	3	985/211
T5	EM	F	56	Ph.D.	29	2	/
T6	EM	M	36	Ph.D.	6	6	985/211

**Table 2 ijerph-20-03276-t002:** Sample data coding.

Research Questions	Opening Coding	Axial Coding	Selective Coding
1. Do Chinese teachers of English experience any emotional challenges in their interactions with ecological systems when addressing students’ plagiarism in the context of academic writing?	I am **frustrated** by the need to reemphasize academic integrity in the class.	The negative emotions experienced by teachers in the context of coping with students’ textual appropriation.	Microsystem
	I feel **very angry** regarding my colleagues’ lenient treatment of plagiarism.		
	I experience **self-condemnatio**n when addressing student plagiarism.		
2-1. Do any emotional changes occur during these interactions?	Bad text copying is a serious issue… I felt **sad** about the students’ poor writing ethics. (Tx-journal 1)Some work was turned in with little revision. I felt **offended**. (Tx-journal 2)	Teachers’ negotiations with students became more tense.	Teachers’ emotions became more negative.
2-2. If so, what factors affect the emotional changes that occur in teachers when viewed from an ecological perspective?	Without **Turnitin**, both teachers and students hardly noticed bad text borrowing. I was sort of **discouraged**.	Lack of plagiarism detection software	University support

## Data Availability

The data presented in this study are available on request from the corresponding author.

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
