# Peer review of "Like Riding a Roller Coaster: University Teachers’ Emotional Experiences Dealing with Student Plagiarism"

_ijerph, 2023, doi:10.3390/ijerph20043276_

Round 1

Reviewer 1 Report

- In the abstract, the data analysis method should be added.

- If the researcher can write the research design process as a schematic diagram, it will make the reader clearer and easier to understand.

- Essay scoring Is there a stand with multiple scorers?

- The researcher should clearly distinguish this research recommendation so the reader can use it for further study. And in what dimensions can the results of this research be used?

Reviewer 2 Report

This manuscript is clear and relevant for the field since it addresses an important issue in current universities, such as plagiarism, from the point of view of the emotional experiences of teachers who face plagiarism from students in their daily practice. References are recent and relevant. The qualitative design of the study seems appropriate since it deals with teachers emotions, its characteristics, associated factors and development. Although as it is noticed in the manuscript, the generalization of the results may be difficult due to the small sample (n=6) and intentional selection. However, the type of study may allow such limitation. The manuscript is described in sufficient detail as to be replicated. The data and tables/figures are easy to understand. The conclusions of the study seem too specific and they could be made more relevant by deeper literature comparison. The study lack implications of the results for better tackling plagiarism and associated teachers' emotions. Ethics statements are adequate.
